# A Quadrature Rule combining
# Control Variates and Adaptive Importance Sampling

**Rémi Leluc**
LTCI, Télécom Paris
Institut Polytechnique de Paris, France
`remi.leluc@telecom-paris.fr`

**François Portier**
CREST
ENSAI, France
`francois.portier@gmail.com`

**Aigerim Zhuman**
LIDAM, ISBA
UCLouvain, Belgium
`aigerim.zhuman@uclouvain.be`

**Johan Segers**
LIDAM, ISBA
UCLouvain, Belgium
`johan.segers@uclouvain.be`

## Abstract

Driven by several successful applications such as in stochastic gradient descent or in Bayesian computation, control variates have become a major tool for Monte Carlo integration. However, standard methods do not allow the distribution of the particles to evolve during the algorithm, as is the case in sequential simulation methods. Within the standard adaptive importance sampling framework, a simple weighted least squares approach is proposed to improve the procedure with control variates. The procedure takes the form of a quadrature rule with adapted quadrature weights to reflect the information brought in by the control variates. The quadrature points and weights do not depend on the integrand, a computational advantage in case of multiple integrands. Moreover, the target density needs to be known only up to a multiplicative constant. Our main result is a non-asymptotic bound on the probabilistic error of the procedure. The bound proves that for improving the estimate's accuracy, the benefits from adaptive importance sampling and control variates can be combined. The good behavior of the method is illustrated empirically on synthetic examples and real-world data for Bayesian linear regression.

## 1   Introduction

In recent years, sequential simulation has emerged as a leading approach to compute multidimensional integrals. A key object in sequential simulation is the sequence of distributions, called the policy, from which to generate the random variables, called particles, used to approximate the integrals of interest. The policy is designed to evolve in the course of the algorithm to mimic the target density, which may itself be known only up to a proportionality constant. While the design of algorithms with adaptive policies has been of major interest recently, only a few studies have focused on using control variates to reduce the variance. This paper provides a new method to incorporate control variates within standard sequential algorithms. The proposed approach significantly improves the accuracy of the initial algorithm, both theoretically and in practice.

**The sequential framework.**   Consider the problem of approximating the integral $\int gf \, \mathrm{d}\lambda = \int_{\mathbb{R}^d} g(x) f(x) \, \mathrm{d}x$, where $\lambda$ is the $d$-dimensional Lebesgue measure, $f$ is a probability density on $\mathbb{R}^d$ and the integrand $g$ is a real-valued function on $\mathbb{R}^d$. For instance, one may think of $f$ as the posterior density in Bayesian inference. Let $(q_i)_{i \geq 0}$ be the policy of the algorithm, i.e., a sequence of probability densities which evolves adaptively depending on previous outcomes. The particles $(X_i)_{i \geq 1}$ are generated sequentially—at iteration $i$, particle $X_i$ is drawn from $q_{i-1}$. The integral $\int gf \, \mathrm{d}\lambda$ is

36th Conference on Neural Information Processing Systems (NeurIPS 2022).

estimated by the normalized sum $\left(\sum_{i=1}^{n} w_i g(X_i)\right)/\left(\sum_{i=1}^{n} w_i\right)$, where $w_i = f(X_i)/q_{i-1}(X_i)$ are the importance weights. The normalization $\sum_{i=1}^{n} w_i$ allows to deal with situations where the target density $f$ is known only up to a proportionality constant.

Such an algorithm is part of the *adaptive importance sampling* (AIS) framework. Many different ways have been investigated to update the densities $q_i$ adaptively. Early works that inspired such sequential schemes include [13, 22, 29] where the sampling policy is chosen out of a parametric family. The parametric approach has been further extended by the Population Monte Carlo framework [4, 5, 26]. Various asymptotic results have been obtained in [6, 10, 34]. In [7, 9, 23, 39], *nonparametric importance sampling* based on kernel smoothing is studied. The latter bears resemblance to *sequential Monte Carlo* methods [8, 6], in which the target distribution $f$ changes in the course of the algorithm.

**Control variates.** Let $h = (h_1, \ldots, h_m)^\top$ be a vector of real-valued functions on $\mathbb{R}^d$ such that for each $k$, the integral $\int h_k f \, \mathrm{d}\lambda$ is known. Without loss of generality, suppose that $\int h f \, \mathrm{d}\lambda = 0$. The functions $h_k$ are called control variates and can be obtained in different ways. In Bayesian statistics, Stein control variates [28] are constructed by applying the second-order Stein operator to functions satisfying certain regularity conditions [27]. Other control variates might be created by re-weighting a function $h^*$ that satisfies $\int h^* \, \mathrm{d}\lambda = 0$ via $h = h^*/f$. The use of control variates is a well studied variance-reduction technique [14, 30]. The benefits can be established theoretically in terms of error bounds [28, 24], weak convergence [35], the excess risk [2] and even uniform error bounds over large classes of integrands [33]. In practice, the control variates framework has led to efficient procedures in reinforcement learning [18, 25] and optimization [38], to name a few. Importance sampling and control variates in case of a Gaussian target density is explored in [20]. The procedure in [21] incorporates control variates and is said to involve adaptive importance sampling, but in fact the particles are always sampled from the uniform distribution on the unit cube. To the best of our knowledge, the existing control variate methods do not account for sequential changes in the particle distribution as is the case in AIS.

**AISCV estimate.** The proposed approach to use control variates within the sequential AIS framework relies on the ordinary least squares expression of control variates (see for instance [35]). To take care of the policy changes, some re-weighting must be applied. The AISCV estimate of the integral $\int g f \, \mathrm{d}\lambda$ is defined as the first coordinate of the solution to the weighted least squares problem

$$(\hat{\alpha}_n, \hat{\beta}_n) = \underset{a \in \mathbb{R}, b \in \mathbb{R}^m}{\arg\min} \sum_{i=1}^{n} w_i \left(g(X_i) - a - b^\top h(X_i)\right)^2,$$

with $w_i$ the importance weights from before. The AISCV estimate $\hat{\alpha}_n$ has several interesting properties: (a) whenever $g$ is of the form $\alpha + \beta^\top h$ for some $\alpha \in \mathbb{R}$ and $\beta \in \mathbb{R}^m$, the error is zero, i.e., $\hat{\alpha}_n = \alpha = \int g f \, \mathrm{d}\lambda$; (b) the estimate takes the form of a quadrature rule $\hat{\alpha}_n = \sum_{i=1}^{n} v_{n,i} g(X_i)$, for quadrature weights $v_{n,i}$ that do not depend on the function $g$ and that can be computed by a single weighted least squares procedure; and (c) it can be computed even when $f$ is known only up to a multiplicative constant. Point (a) suggests that when the linear combinations of the functions $h_k$ span a rich function class, the integration error is likely to be small. Point (b) implies that multiple integrals can be computed just as easily as a single one. Point (c) shows that the approach is applicable for Bayesian computations. In addition, the control variates can be brought into play in a *post-hoc* scheme, after generation of the particles and importance weights, and this for any AIS algorithm.

**Main result.** The main theoretical result of the paper is a probabilistic, non-asymptotic bound on $\hat{\alpha}_n - \alpha$. Under appropriate conditions, the bound scales as $\tau/\sqrt{n}$, where $\tau^2$ is the scale constant in a sub-Gaussian tail condition on the error variable $\varepsilon = g - \alpha - \beta^\top h$ for $(\alpha, \beta) = \arg\min_{a,b} \int (g - a - b^\top h)^2 f \, \mathrm{d}\lambda$. Note that $\varepsilon$ has the smallest possible variance one could get using control variates $h$. As a consequence, when the space of control variates is well suited for approximating $g$, the AISCV estimate will be highly accurate. Also, our bound depends only on the linear function space spanned by the control variates $h_1, \ldots, h_m$, not on the particular basis chosen in that space. The results rely on martingale theory, in particular on a concentration inequality for norm-subGaussian martingales in [19]. In the course of the proof, we develop a novel bound on the smallest eigenvalue of certain random matrices, extending an inequality from [37] to the martingale case.

**Outline.** Section 2 introduces the general framework of adaptive importance sampling and control variates. Next, Section 3 presents the AISCV estimate and the associated quadrature rule. Section 4 contains the statements of the theoretical results while Section 5 gathers practical considerations, including the construction of control variates. Numerical experiments are presented in Section 6.

## 2 Preliminaries on Monte Carlo integration

The aim of this section is to present the required mathematical framework for Monte Carlo integration and the variance reduction methods of interest, namely adaptive importance sampling and the control variate technique. Recall that $g : \mathbb{R}^d \to \mathbb{R}$ is an integrand and $f$ a probability density on $\mathbb{R}^d$. The aim is to compute $\mathbb{E}_f[g] = \int gf \, \mathrm{d}\lambda$.

**Adaptive importance sampling.** In adaptive importance sampling (AIS), $\mathbb{E}_f[g]$ is estimated by a weighted mean over a sample of random particles $X_1, \ldots, X_n$ in $\mathbb{R}^d$. Since appropriate sampling densities naturally depend on $g$ and $f$, we generally cannot simulate from them. They are then approximated in an adaptive manner by a family of tractable densities $(q_i)_{i \geq 0}$ that often evolve towards a density $q_{\text{opt}}$ that optimizes some criterion. While the starting density $q_0$ is fixed, the density $q_i$ for $i \geq 1$ is determined in function of the particles $X_1, \ldots, X_i$ already sampled; think for instance of a parametric family, where the parameter of $q_i$ is a function of $X_1, \ldots, X_i$. Given the particles $X_1, \ldots, X_i$, the next particle, $X_{i+1}$, is then drawn from $q_i$. Formally, let $(X_i)_{i \geq 1}$ be a sequence of random vectors on $\mathbb{R}^d$ defined on some probability space $(\Omega, \mathscr{F}, \mathbb{P})$. The distribution of the sequence $(X_i)_{i \geq 1}$ is specified by its policy as defined below.

**Definition 1** (Policy). *A policy is a random sequence of probability density functions $(q_i)_{i \geq 0}$ on $\mathbb{R}^d$ adapted to the $\sigma$-field $(\mathscr{F}_i)_{i \geq 0}$ defined by $\mathscr{F}_0 = \{\varnothing, \Omega\}$ and $\mathscr{F}_i = \sigma(X_1, \ldots, X_i)$ for $i \geq 1$. The sequence $(q_i)_{i \geq 0}$ is the policy of $(X_i)_{i \geq 1}$ whenever $X_i$ has density $q_{i-1}$ conditionally on $\mathscr{F}_{i-1}$.*

The (normalized) adaptive importance sampling estimate of $\mathbb{E}_f[g]$ is then defined as

$$I_n^{(\text{ais})}(g) = \frac{\sum_{i=1}^n w_i g(X_i)}{\sum_{i=1}^n w_i} \quad \text{where} \quad w_i = \frac{f(X_i)}{q_{i-1}(X_i)} \quad \text{for } i = 1, \ldots, n. \tag{1}$$

The sampling weights $w_i$ reflect the fact that $X_i$ has been sampled from $q_{i-1}$ rather than from $f$. The division by $\sum_{i=1}^n w_i$ rather than by $n$ has two benefits: first, the integration is exact for constant integrands, and second, $f$ needs to be known only up to a proportionality constant, an advantage for Bayesian inference.

Since updating the density $q_i$ at each iteration may be computationally expensive, it is customary to hold it fixed over a pre-determined number of iterations. Writing $n = n_1 + \cdots + n_T$ in terms of positive integers $(n_t)_{t=1}^T$ called the *allocation policy*, the AIS estimate then becomes

$$I_T^{(\text{ais})}(g) = \frac{\sum_{t=1}^T \sum_{i=1}^{n_t} w_{t,i} g(X_{t,i})}{\sum_{t=1}^T \sum_{i=1}^{n_t} w_{t,i}} \quad \text{where} \quad w_{t,i} = \frac{f(X_{t,i})}{q_t(X_{t,i})} \tag{2}$$

for $t = 1, \ldots, T$ and $i = 1, \ldots, n_t$. At stage $t$, the particles $X_{t,1}, \ldots, X_{t,n_t}$ are sampled independently from $q_{t-1}$, while all particles sampled up to and including stage $t$ are used to determine the sampling density $q_t$ for stage $t + 1$. It is easy to see that the two formulations of the AIS estimate are equivalent: (1) arises from (2) by setting $n_t = 1$ for all $t$, while (2) can be obtained from (1) by constructing the policy in such a way that the densities $q_i$ do not change within integer intervals of the form $\{0, \ldots, n_1 - 1\}$, $\{n_1, \ldots, n_1 + n_2 - 1\}$, and so on. While the shorter representation (1) is more convenient for theoretical purposes, formulation (2) is the one used in practice (see Section 6).

Interestingly, the AIS estimate (1) may be seen as a weighted least-squares estimate minimizing the loss function $a \mapsto \sum_{i=1}^n w_i (g(X_i) - a)^2$. This perspective is key to understand control variates.

**Control variates.** The control variates method is a variance reduction technique that consists in incorporating a new piece of information—the known values of the integrals of some control functions—in a basic Monte Carlo framework. Control variates are simply functions $h_1, \ldots, h_m \in L_2(f)$ with known integrals. Without loss of generality, assume that $\mathbb{E}_f[h_j] = 0$ for all $j = 1, \ldots, m$. Let $h = (h_1, \ldots, h_m)^\top$ denote the $\mathbb{R}^m$-valued function with the $m$ control variates as elements. For any coefficient vector $\beta \in \mathbb{R}^m$, we have $\mathbb{E}_f[g - \beta^\top h] = \mathbb{E}_f[g]$. Given an independent random sample $X_1, \ldots, X_n$ from $f$, any $\beta \in \mathbb{R}^m$ therefore results in an unbiased estimator of $\mathbb{E}_f[g]$ by

$$I_n^{(\text{cv})}(g, \beta) = \frac{1}{n} \sum_{i=1}^n \left( g(X_i) - \beta^\top h(X_i) \right). \tag{3}$$

Provided the $m \times m$ covariance matrix $G = \mathbb{E}_f[hh^\top]$ is invertible, there is a unique coefficient vector $\beta^* \in \mathbb{R}^m$ for which the variance of $I_n^{(\text{cv})}(g)$ is minimal and it is given by

$$\beta^* = \left( \mathbb{E}_f[hh^\top] \right)^{-1} \mathbb{E}_f[hg]. \tag{4}$$

This vector being generally unknown, it needs to be estimated from the particles $X_1, \ldots, X_n$. Casting the problem in an ordinary least squares framework leads to the control variate estimate

$$I_n^{(\mathrm{cv})}(g) = I_n^{(\mathrm{cv})}\big(g, \hat{\beta}_n^{(\mathrm{cv})}\big) = \hat{\alpha}_n^{(\mathrm{cv})} \qquad \text{where}$$

$$\big(\hat{\alpha}_n^{(\mathrm{cv})}, \hat{\beta}_n^{(\mathrm{cv})}\big) \in \operatorname*{arg\,min}_{(a,b)\in\mathbb{R}\times\mathbb{R}^m} \frac{1}{n} \sum_{i=1}^n \big(g(X_i) - a - b^\top h(X_i)\big)^2. \qquad (5)$$

The estimator $I_n^{(\mathrm{cv})}(g)$ is well-defined provided the minimizer $\hat{\alpha}_n^{(\mathrm{cv})}$ to (5) is unique. This is the case if and only if there does not exist $b \in \mathbb{R}^m$ such that $b^\top h(X_i) = 1$ for all $i = 1, \ldots, n$.

The asymptotic distribution of $I_n^{(\mathrm{cv})}(g)$ as $n \to \infty$ is the same as if the variance-minimizing vector $\beta^*$ were used in (3). In particular, the asymptotic variance of $I_n^{(\mathrm{cv})}(g)$ is $\sigma_m^2(g)/n$ where

$$\sigma_m^2(g) = \min_{\beta\in\mathbb{R}^m} \mathbb{E}_f\big[(g - \mathbb{E}_f[g] - \beta^\top h)^2\big].$$

Interestingly, when using only the first $\ell$ out of $m$ control variates, where $\ell \in \{0, 1, \ldots, m\}$, we have $\sigma_m^2(g) \le \sigma_\ell^2(g)$. In terms of asymptotic variance, it therefore never harms to add more control variates. Their construction will be addressed in Section 5.1.

## 3 Combining adaptive importance sampling with control variates

**AISCV estimator.** Consider the same integration problem $\mathbb{E}_f[g] = \int gf \, \mathrm{d}\lambda$ as in Section 2. With the idea of performing variance reduction when calculating integrals with respect to the posterior density in Bayesian inference, we incorporate control variates into the AIS estimate. Let the particles $(X_i)_{i\ge1}$ be generated according to a policy $(q_i)_{i\ge0}$ as in Definition 1. Let $h = (h_1, \ldots, h_m)^\top$ be a vector of control variates, i.e., $h_j \in L_2(f)$ and $\mathbb{E}_f[h_j] = 0$ for every $j = 1, \ldots, m$. Combining (1) and (3), the proposed estimate takes the form

$$I_n^{(\mathrm{aiscv})}(g, \beta) = \frac{\sum_{i=1}^n w_i \big(g(X_i) - \beta^\top h(X_i)\big)}{\sum_{i=1}^n w_i}, \qquad (6)$$

where $\beta \in \mathbb{R}^m$ remains to be determined. To do so, the ordinary least-squares problem in (5) is replaced by a weighted one, yielding the novel AISCV estimator

$$I_n^{(\mathrm{aiscv})}(g) = I_n^{(\mathrm{aiscv})}\big(g, \hat{\beta}_n\big) = \hat{\alpha}_n \qquad \text{where}$$

$$\big(\hat{\alpha}_n, \hat{\beta}_n\big) \in \operatorname*{arg\,min}_{(a,b)\in\mathbb{R}\times\mathbb{R}^m} \sum_{i=1}^n w_i \big(g(X_i) - a - b^\top h(X_i)\big)^2. \qquad (7)$$

The estimator is well-defined only if the minimizer $\hat{\alpha}_n$ is unique—the minimizer $\hat{\beta}_n$ need not be. We will come back to this in the next paragraph.

As in (2), the policy may be divided into $T$ stages in order to reduce the number of times the sampler needs to be updated. Stage $t = 1, \ldots, T$ has length $n_t$, with $\sum_{t=1}^T n_t = n$. Within each stage, the sampling density remains constant. In practice, this leads to the AISCV estimate in Algorithm 1.

**Quadrature rule.** The AIS estimate (1) is a quadrature rule with quadrature points $X_i$ and quadrature weights proportional to the sampling weights $w_i$. The AISCV estimate (7) has the same property, but with adapted quadrature weights. Let $e_n = (e_{n,i})_{i=1,\ldots,n}$ be the vector of residuals resulting from the weighted least-squares regression of the constant vector $\mathbf{1}_n = (1, \ldots, 1)^\top \in \mathbb{R}^n$ on the control variates but without intercept:

$$e_{n,i} = 1 - \hat{\beta}_n(\mathbf{1}_n)^\top h(X_i) \qquad \text{where}$$

$$\hat{\beta}_n(\mathbf{1}_n) \in \operatorname*{arg\,min}_{b\in\mathbb{R}^m} \sum_{i=1}^n w_i \big(1 - b^\top h(X_i)\big)^2. \qquad (8)$$

Even though the vector $\hat{\beta}_n(\mathbf{1}_n)$ is not necessarily unique, the weighted least squares fit $(\hat{\beta}_n(\mathbf{1}_n)^\top h(X_i))_{i=1,\ldots,n}$ always is. According to the next proposition, the quadrature weights are proportional to $(w_i e_{n,i})_{i=1,\ldots,n}$.

---

**Algorithm 1** Adaptive Importance Sampling with Control Variates (AISCV)

---

**Require:** integrand $g$, target density $f$ (up to a proportionality constant), number of stages $T \in \mathbb{N}^*$, allocation policy $(n_t)_{t=1}^T$, initial density $q_0$, update rule for the sampling policy

1: **for** $t = 1, \ldots, T$ **do**
2:      Generate an independent random sample $X_{t,1}, \ldots, X_{t,n_t}$ from $q_{t-1}$
3:      Compute the vector of weights $(w_{t,i})_{i=1}^{n_t}$ where $w_{t,i} = f(X_{t,i})/q_{t-1}(X_{t,i})$
4:      Construct the matrix of control variates $H_t = \left( h_j(X_{t,i}) \right)_{i=1,\ldots,n_t}^{j=1,\ldots,m}$
5:      Evaluate the integrand in the particles: $(g(X_{t,i}))_{i=1}^{n_t}$
6:      Update the sampler $q_t$ based on all previous particles $(X_{s,i} : s = 1, \ldots, t; i = 1, \ldots, n_s)$
7: **end for**
8: Compute $(\hat{\alpha}_T, \hat{\beta}_T) = \arg\min_{(a,b) \in \mathbb{R} \times \mathbb{R}^m} \left\{ \sum_{t=1}^T \sum_{i=1}^{n_t} w_{t,i} \left( g(X_{t,i}) - a - b^\top h(X_{t,i}) \right)^2 \right\}$
9: **return** $I_n^{(\mathrm{aiscv})}(g) = \hat{\alpha}_T$.

---

**Proposition 1** (AISCV quadrature rule). *The minimizer $\hat{\alpha}_n$ in (7) is unique if and only if $e_n \neq 0$ in (8). In that case, the AISCV estimate is*

$$I_n^{(\mathrm{aiscv})}(g) = \hat{\alpha}_n = \frac{\sum_{i=1}^n w_i e_{n,i} g(X_i)}{\sum_{i=1}^n w_i e_{n,i}}. \tag{9}$$

If $e_n = 0$, then there exists $b \in \mathbb{R}^m$ such that $b^\top h(X_i) = 1$ for all $i = 1, \ldots, n$. In that case, the minimizer $\hat{\alpha}_n$ in (7) is not unique and the AISCV estimate is not well-defined. To remedy this, one can for instance reduce the number of control variates. This issue already occurs with the ordinary control variate estimator in (3).

Rather than requiring a different weighted least squares problem for every integrand $g$ as in (7), the quadrature rule in (9) only involves a single weighted least squares problem (8), whatever $g$. Given the quadrature weights, calculating the AISCV estimate for a novel integrand only requires the evaluations of that function on the sampled particles, making the whole procedure a *post-hoc* scheme. The steps in case the sampling policy is divided into $T$ stages are given in Algorithm 2, which gives the same result as Algorithm 1, but with less effort if multiple integrands $g$ are into play.

---

**Algorithm 2** Quadrature Rule – AISCV *post-hoc* scheme

---

**Require:** integrand $g$, $T \in \mathbb{N}^*$, allocation policy $(n_t)_{t=1}^T$, weights $(w_t)_{t=1}^T$ with $w_t = (w_{t,i})_{i=1}^{n_t}$, matrices $(H_t)_{t=1}^T$ with $H_t = \left( h_j(X_{t,i}) \right)_{i=1,\ldots,n_t}^{j=1,\ldots,m}$, particles $(X_{t,i} : t = 1, \ldots, T; i = 1, \ldots, n_t)$

1: Compute $\hat{\beta}_n(\mathbf{1}_n) = \arg\min_{b \in \mathbb{R}^m} \sum_{t=1}^T \sum_{i=1}^{n_t} w_{t,i} \left( 1 - b^\top h(X_{t,i}) \right)^2$
2: Compute $u_t = \mathrm{diag}(w_t)[\mathbf{1}_{n_t} - H_t \hat{\beta}_n(\mathbf{1}_n)]$ for $t = 1, \ldots, T$
3: Compute $s = \sum_{t=1}^T \sum_{i=1}^{n_t} u_{t,i}$
4: Compute weights $v_{t,i} = u_{t,i}/s$ for $t = 1, \ldots, T$ and $i = 1, \ldots, n_t$
5: **return** $I_T^{(\mathrm{aiscv})}(g) = \sum_{t=1}^T \sum_{i=1}^{n_t} v_{t,i} g(X_{t,i})$

---

## 4   Theoretical properties of the AISCV estimate

Here we point out several theoretical properties of the novel AISCV estimate. A first point is that the integration rule is exact on the linear span of the control variates and the constant function.

**Proposition 2** (Exact integration). *For integrands of the form $g = \alpha + \beta^\top h$ for $\alpha \in \mathbb{R}$ and $\beta \in \mathbb{R}^m$, the AISCV estimate is exact: $I_n^{(\mathrm{aiscv})}(g) = \alpha = \mathbb{E}_f[g]$.*

A second property is that we may apply arbitrary invertible linear transformations to the control variates without changing the AISCV estimate. This can be advantageous computationally, to make the underlying weighted least squares problem more stable numerically. Also, it means that without

loss of generality, we may assume that the control variates are uncorrelated and have unit variance, which simplifies the theoretical performance analysis.

**Proposition 3** (Invariance). *If the matrix $A \in \mathbb{R}^{m \times m}$ is invertible, then the AISCV estimate based on the control variates $Ah$ is the same as the one based on $h$.*

Our main result is a non-asymptotic bound on the error of the AISCV estimate for $\int gf \, d\lambda$ when $\int g^2 f \, d\lambda$ is finite. First, we introduce some assumptions and definitions.

The first condition that is required concerns the policy given by the AIS part of the algorithm. It is supposed that any element from the policy should dominate the function $f$.

**Assumption 1** (Dominated measures). *There exists $c \geq 1$ such that, for all $x \in \mathbb{R}^d$ and for any $i = 1, \ldots, n$, we have $f(x) \leq c \cdot q_i(x)$.*

This assumption represents a *safe* approach to importance sampling, as the policy will always allow to sample in places where $f$ is positive. A well-known and well-spread [16, 30, 9] technique to achieve such a defensive strategy is to a use mixture density $q_i = (1 - \eta)f_i + \eta q_0$ where $\eta \in (0, 1)$ and where $q_0$ has sufficiently heavy tails to dominate $f$. Such a mixture allows to choose the densities $f_i$ with some flexibility using in principle any AIS algorithm. Second, the control variates shall be linearly independent and bounded.

**Assumption 2** (Control variates). *We have $\sup_{x:f(x)>0} |h_j(x)| < \infty$ for all $j = 1, \ldots, m$. The matrix $G = \int hh^\top f \, d\lambda$ is invertible.*

The previous condition allows to define the standardized vector of control variates as $\hbar = G^{-1/2}h$. By Proposition 3, this change does not affect the AISCV estimate. The orthonormal control variates $\hbar$ will play a key role through the following quantity

$$B = \sup_{x:f(x)>0} \|\hbar(x)\|_2^2.$$

The quadratic form $\|\hbar(x)\|_2^2 = h(x)^\top G^{-1} h(x)$ is referred to as the *leverage function* in ordinary linear regression as it quantifies the influence of a training point $x$ on the prediction of the observed response. It is invariant with respect to invertible linear transformations of the control variate vector.

Assumption 2 and the fact that the integrand $g$ is square integrable with respect to $f$ allows to define the residual function $\varepsilon = g - \int gf \, d\lambda - h^\top \beta^*$ where $\beta^*$ has been introduced in (4) as a minimizer of the residual variance. Since we work in the space $L^2(f)$, we assume without loss of generality that $g$ and $h$ vanish outside $\{x : f(x) > 0\}$ and we put $\varepsilon(x) = 0$ for $x \in \mathbb{R}^d$ such that $f(x) = 0$. The residual function $\varepsilon$ should satisfy the following tail condition.

**Assumption 3** (Residual tail). *There exists $\tau > 0$ such that, for all $t > 0$ and all integer $i \geq 1$, we have $\mathbb{P}[|w_i\varepsilon(X_i)| > t \mid \mathscr{F}_{i-1}] \leq 2\exp(-t^2/(2\tau^2))$.*

The previous assumption concerns both the function $\varepsilon$ and the policy sequence $(q_i)_{i \geq 0}$. Since $\mathbb{E}[w_i\varepsilon(X_i) \mid \mathscr{F}_{i-1}] = 0$, it is implied by the so-called sub-Gaussian condition [3] that $\mathbb{E}[\exp(\lambda w_i\varepsilon(X_i)) \mid \mathscr{F}_{i-1}] \leq \exp(-\lambda^2\tau^2/2)$ for any $\lambda \in \mathbb{R}$. In the proof of Theorem 1, Assumption 3 allows to derive concentration bounds on residual-based sums using recent results from [19, 24]. We are now in position to state our main result on the error of the AISCV estimate.

**Theorem 1** (Concentration inequality for AISCV estimate). *If Assumptions 1, 2 and 3 hold, then, for any $\delta \in (0, 1)$ and for all $n \geq C_1 c^2 B \log(10m/\delta)$, we have, with probability at least $1 - \delta$, that*

$$\left| I_n^{(\mathrm{aiscv})}(g) - \int_{\mathbb{R}^d} g(x)f(x) \, dx \right| \leq C_2\tau\sqrt{\frac{\log(10/\delta)}{n}} + C_3 cB\tau\frac{\log(10m/\delta)}{n},$$

*where $C_1$, $C_2$, $C_2$ are universal constants specified in the proof.*

**Remark 1** (Understanding $\tau$). *The quantity $\tau$ in Assumption 3 is related to the conditional variance $\mathbb{E}[w_i^2\varepsilon^2(X_i) \mid \mathscr{F}_{i-1}]$. They actually coincide when $w_i\varepsilon(X_i)$ is Gaussian. For a policy satisfying Assumption 1, $\mathbb{E}[w_i^2\varepsilon^2(X_i) \mid \mathscr{F}_{i-1}] \leq c\sigma_m^2$ which for certain combinations of integrands and control functions scales as $m^{-s/d}$ [35] where the parameter $s$ represents the degree of smoothness of $g$.*

**Remark 2** (Convergence rates). *Consider an asymptotic regime where the number of control variates $m$ tends to infinity with the sample size $n$. The AISCV estimate improves upon the AIS method $(m = 0)$, which has rate $1/\sqrt{n}$, as soon as $\tau + \tau B \log(m)/\sqrt{n} \to 0$. To recover the same order of an oracle estimate with rate $\tau/\sqrt{n}$, one must have $B \log(m) = O(\sqrt{n})$ as $n \to \infty$.*

# 5 Practical considerations

This section presents ways to build control variates using either families of polynomials or general functions based on Stein's method, with a highlight on computations in the Bayesian framework.

## 5.1 Control variate constructions

**Orthogonal polynomials.** When the target density $f$ can be decomposed as a product of univariate densities $f = p_1 \otimes \cdots \otimes p_d$, multidimensional control functions may be constructed based on univariate ones. This happens for instance for the uniform distribution over the unit cube $[0, 1]^d$ or with uncorrelated Gaussian distributions on $\mathbb{R}^d$. Such univariate control variates may be easily constructed using families of polynomials [12], such as Legendre polynomials for the uniform distribution on $[0, 1]$ and Hermite polynomials for the Gaussian distribution on $\mathbb{R}$. This technique can also be used when $f$ is dominated by another density $f^*$ having the said product form by transforming zero-mean control variates $h^*$ with respect to $f^*$ via $h = h^* f^*/f$.

Let $(h_1, \ldots, h_k)$ be a vector of univariate control functions with respect to a density $p$, i.e., $\mathbb{E}_p[h_j] = 0$ for all $j = 1, \ldots, k$. Let $h_0 = 1$ denote the constant function equal to one. For a multi-index $\ell = (\ell_1, \ldots, \ell_d)$ in $\{0, \ldots, k\}^d \setminus \{(0, \ldots, 0)\}$, multivariate controls with respect to $p^{\otimes d}$ are built by forming tensor products of the form $h_\ell(x_1, \ldots, x_d) = h_{\ell_1}(x_1) \cdots h_{\ell_d}(x_d)$, yielding a total number of $m = (k + 1)^d - 1$ control functions. Alternative approaches yielding smaller control spaces consist of imposing $\ell_j = 0$ for all but a small number of coordinates $j = 1, \ldots, d$ or by the constraint $\ell_1 + \cdots + \ell_d \leq Q$ for some $Q \geq 1$.

**Stein control variates.** In the general case where one has only access to the evaluations of $f$, control variates may be constructed using Stein's method. The technique relies on the gradient $\nabla_x \log f(x)$ which can either be directly computed (see the example of Bayesian regression below) or which may be available through automatic differentiation provided in popular API's such as Tensorflow and PyTorch [1, 31]. Let $\Delta_x = \nabla_x^\top \nabla_x$ denote the Laplace operator. By definition, the second-order Stein operator $\mathscr{L}$ [36, 15] associated to the density $f$ is defined by:

$$\forall \varphi \in \mathscr{C}^2(\mathbb{R}^d, \mathbb{R}), \quad (\mathscr{L}\varphi)(x) = \Delta_x \varphi(x) + \nabla_x \varphi(x)^\top \nabla_x \log f(x).$$

The transformation guarantees that $\mathbb{E}_f[\mathscr{L}\varphi] = 0$ for all $\varphi$ with weak regularity conditions [27]. Therefore, we can build infinitely many control variates $h_\varphi = \mathscr{L}\varphi$ from given functions $\varphi$. One simple way is to let $\varphi$ be a polynomial with bounded total degree: for a degree vector $\boldsymbol{\alpha} = (\alpha_1, \ldots, \alpha_d) \in \mathbb{N}^d$ with $\alpha_1 + \cdots + \alpha_d \leq Q$, define $\varphi_{\boldsymbol{\alpha}}(x) = x_1^{\alpha_1} \cdots x_d^{\alpha_d}$. Given the dimension $d$ and the total degree $Q$, there are $m = \binom{d+Q}{d} - 1$ such degree vectors, yielding the associated control variates $h_{\boldsymbol{\alpha}} = h_{\varphi_{\boldsymbol{\alpha}}}$. For fast computation, note that, writing $\phi_{\boldsymbol{\alpha}}(x) = \varphi_{\boldsymbol{\alpha}}(x)\mathbb{1}_d$, $D_1(x) = \text{diag}(\alpha_1/x_1, \ldots, \alpha_d/x_d)$ and $D_2(x) = \text{diag}(\alpha_1(\alpha_1 - 1)/x_1^2, \ldots, \alpha_d(\alpha_d - 1)/x_d^2)$, we have $\nabla_x \varphi_{\boldsymbol{\alpha}}(x) = D_1(x)\phi_{\boldsymbol{\alpha}}(x)$ and $\Delta_x \varphi_{\boldsymbol{\alpha}}(x) = \mathbb{1}_d^\top(D_2(x)\phi_{\boldsymbol{\alpha}}(x))$. In practice, all combinations of $\boldsymbol{\alpha}$ are stored in a matrix $A \in \mathbb{N}^{m \times d}$.

## 5.2 Bayesian inference

Given data $\mathscr{D}$ and a parameter of interest $\theta \in \Theta \subset \mathbb{R}^d$, posterior integrals take the form $\int_{\mathbb{R}^d} g(\theta) p(\theta|\mathscr{D}) \, d\theta$, where $p(\theta|\mathscr{D}) \propto \ell(\mathscr{D}|\theta)\pi(\theta)$ is the posterior distribution, proportional to a prior $\pi(\theta)$ and a likelihood function $\ell(\mathscr{D}|\theta)$. For instance, when $g(\theta) = \theta$, the integral above recovers the posterior mean. Stein control variates involve the computation of the gradient of the log-posterior $\nabla_\theta \log p(\theta|\mathscr{D})$, which implicitly relies on the score function $\nabla_\theta \log \ell(\mathscr{D}|\theta)$. We point out two common examples—linear and logistic regression—where these functions are easy to compute.

**Bayesian linear regression.** Consider a linear regression problem comprised of observations $X \in \mathbb{R}^{N \times d}$ with labels $y \in \mathbb{R}^N$. In the Gaussian fixed design setting, the predictor $x_i$ produces the response $y_i = x_i^\top \theta + \varepsilon_i$ where $\varepsilon_1, \ldots, \varepsilon_N \sim \mathcal{N}(0, \sigma^2)$ are centered Gaussian noises. The likelihood $\ell(X, y|\theta)$ is proportional to $(\sigma^2)^{-N/2} \exp(-(y - X\theta)^\top(y - X\theta)/(2\sigma^2))$, yielding the score function $\nabla_\theta \log \ell(X, y|\theta) = X^\top(y - X\theta)/(2\sigma^2)$.

**Bayesian logistic regression.** Next, consider the logistic regression problem comprised of observations $X \in \mathbb{R}^{N \times d}$ with associated binary labels $y \in \{0, 1\}^N$. Letting $\sigma(s) = 1/(1 + e^{-s})$ denote the sigmoid function, the likelihood function is $\ell(X, y|\theta) = \prod_{i=1}^N \sigma(\theta^\top x_i)^{y_i}(1 - \sigma(\theta^\top x_i))^{1-y_i}$. The score function is simply $\nabla_\theta \log \ell(X, y|\theta) = X^\top(y - \sigma(X\theta))$.

# 6 Numerical illustration

To compare the finite-sample performance of the AIS and AISCV estimators, we first present in Section 6.1 synthetic data examples involving the integration problem over the unit cube $[0,1]^d$ and then with respect to some Gaussian mixtures as in [4]. The goal is to compute $\int g f \, d\lambda$ for vectors of integrands $g : \mathbb{R}^d \to \mathbb{R}^p$. We consider various dimensions $d > 1$ and several choices for the number of control variates $m$. Section 6.2 deals with real-world datasets in the context of Bayesian inference. For ease of reproducibility, the code, numerical details and additional results are available in the supplementary material.

**Parameters.** In all simulations, the sampling policy is taken within the family of multivariate Student $t$ distributions of degree $\nu$ denoted by $\{q_{\mu,\Sigma_0} : \mu \in \mathbb{R}^d\}$ with $\Sigma_0 = \sigma_0 I_d(\nu - 2)/\nu$ and $\nu > 2, \sigma_0 > 0$. Similarly to [34], the mean $\mu_t$ is updated at each stage $t = 1, \ldots, T$ by the generalized method of moments (GMM), leading to $\mu_t = (\sum_{s=1}^{t} \sum_{i=1}^{n_s} w_{s,i} X_{s,i})/(\sum_{s=1}^{t} \sum_{i=1}^{n_s} w_{s,i})$. The allocation policy is fixed to $n_t = 1000$ and the number of stages is $T \in \{5; 10; 20; 30; 50\}$. The different Monte Carlo estimates are compared by their mean squared error (MSE) obtained over 100 independent replications.

## 6.1 Synthetic examples

**Integration on** $[0,1]^d$**.** We seek to integrate functions $g$ with respect to the uniform density $f(x) = 1$ for $x \in [0,1]^d$ in dimensions $d \in \{4; 8\}$. We rely on Legendre polynomials for the control variates. Consider the integrands $g_1(x) = 1 + \sin\left(\pi(2d^{-1}\sum_{i=1}^{d} x_i - 1)\right)$, $g_2(x) = \prod_{i=1}^{d}(2/\pi)^{1/2} x_i^{-1} e^{-\log(x_i)^2/2}$ and $g_3(x) = \prod_{i=1}^{d} \log(2) 2^{1-x_i}$, all of which integrate to 1 on $[0,1]^d$. None of the integrands is a linear combination of the control variates. The policy parameters are $\mu_0 = (0.5, \ldots, 0.5) \in \mathbb{R}^d$, $\nu = 8$, and $\sigma_0 = 0.1$. The control variates are built out of tensor products of Legendre polynomials where the degree $\ell_j$ equals 0 for all but two coordinates, leading to a total number of $m = kd + k^2 d(d-1)/2$ control variates. The maximum degree in each variable is $k = 6$, yielding $m = 240$ and $m = 1056$ control variates in dimensions $d = 4$ and $d = 8$ respectively. Figure 1 presents the boxplots of the AIS and AISCV estimates. The error reduction obtained thanks to the control variates is huge: the AISCV estimate has a mean squared error smaller than the one of the AIS estimate by a factor at least 10 and up to 100 (see Table 1 in the supplement).

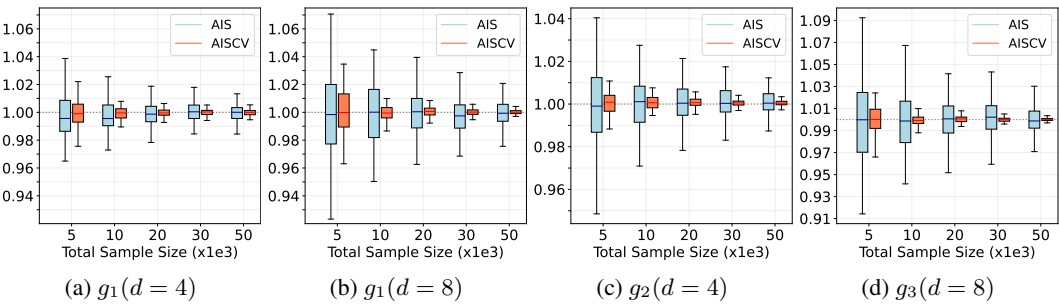

|  (a) $g_1(d=4)$ | (b) $g_1(d=8)$ | (c) $g_2(d=4)$ | (d) $g_3(d=8)$ |

Figure 1: Integration on $[0,1]^d$: boxplots of estimates $I_n^{(\mathrm{ais})}(g)$ and $I_n^{(\mathrm{aiscv})}(g)$ with integrands $g_1, g_2, g_3$ in dimensions $d \in \{4; 8\}$ obtained over 100 replications. The true integral equals 1.

**Gaussian mixture $f$ and Stein control variates.** In this setting we assume we only have access to the evaluations of the target density $f$. We consider the classical example introduced in [4] where $f$ is a mixture of two Gaussian distributions. The control variates are built using Stein's method (Section 5.1) out of polynomials of total degree at most $Q \in \{2; 3\}$, leading to a number of control variates $m \in \{14; 34\}$ in dimension $d = 4$ and $m \in \{44; 164\}$ in dimension $d = 8$ respectively. We consider two cases: an isotropic and an anisotropic one.

*Isotropic case.* Let $f_\Sigma(x) = 0.5\Phi_\Sigma(x - \mu) + 0.5\Phi_\Sigma(x + \mu)$ where $\mu = (1, \ldots, 1)^\top/2\sqrt{d}$, $\Sigma = I_d/d$ and $\Phi_\Sigma$ is the multivariate normal density function with zero mean and covariance matrix $\Sigma$. The Euclidean distance between the two mixture centers is 1, independently of $d$. The initial density $q_0$ is the multivariate Student $t$ distribution with mean $(1, -1, 0, \ldots, 0)/\sqrt{d}$ and variance $(5/d)I_d$. The

initial mean value differs from the null vector to prevent the naive algorithm using the initial density from having good results due to the symmetrical set-up.

*Anisotropic case.* In this case, the mixture is unbalanced and each Gaussian is anisotropic. The target density is $f_V(x) = 0.75\Phi_V(x - \mu) + 0.25\Phi_V(x + \mu)$ where $\mu = (1, \ldots, 1)^\top/2\sqrt{d}$ and $V = \mathrm{diag}(10, 1, \ldots, 1)/d$. The initial density $q_0$ is the same as for the isotropic case.

Figure 2 presents the evolution of the logarithm of the mean squared error $\|\hat{I}(g) - I(g)\|_2^2$. Once again, the AISCV estimators are the clear winners with a mean squared error smaller by a factor up to 1000 for the anisotropic case (see Table 2 in the supplement).

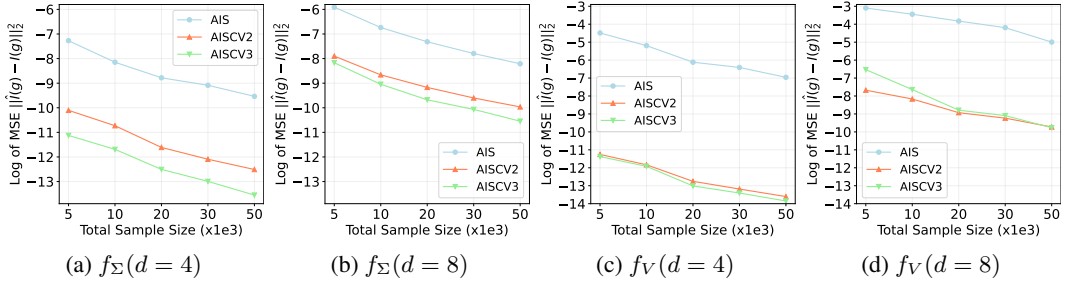

(a) $f_\Sigma(d = 4)$        (b) $f_\Sigma(d = 8)$        (c) $f_V(d = 4)$        (d) $f_V(d = 8)$

Figure 2: Gaussian mixture density: Logarithm of $\|\hat{I}(g) - I(g)\|_2^2$ for $g(x) = x$ with target isotropic $f_\Sigma$ and anisotropic $f_V$ in dimensions $d \in \{4; 8\}$ obtained over 100 replications.

## 6.2   Real-world examples

We place ourselves in the framework of Bayesian linear regression (Section 5.2) with features $X \in \mathbb{R}^{N \times d}$ and continuous responses $y \in \mathbb{R}^N$. The posterior distribution $p(\theta|\mathscr{D})$ involves a Gaussian prior $\pi(\theta) \sim \mathscr{N}(\mu_a, \Sigma_a)$ and a likelihood function $\ell(\mathscr{D}|\theta)$ proportional to $(\sigma^2)^{-N/2} \exp(-(y - X\theta)^\top(y - X\theta)/(2\sigma^2))$. The noise level is fixed and taken sufficiently large at $\sigma = 50$ to account for general priors. The posterior distribution is Gaussian too: $\mathscr{N}(\mu_b, \Sigma_b)$ with $\mu_b = \Sigma_b(\sigma^{-2}X^\top y + \Sigma_a^{-1}\mu_a)$ and $\Sigma_b = (\sigma^{-2}X^\top X + \Sigma_a^{-1})^{-1}$. The integrand is $g(\theta) = \sum_{i=1}^d \theta_i^2$ and the control variates are built with the Stein operator (Section 5.1) out of monomials with total degree $Q \in \{1; 2\}$, leading to the AISCV1 and AISCV2 estimators respectively.

**Datasets and parameters.** Classical datasets from [11] are considered : *housing* ($N = 506; d = 13; m \in \{12; 104\}$); *abalone* ($N = 4177; d = 8; m \in \{7; 44\}$); *red wine* ($N = 1599; d = 11; m \in \{10; 77\}$); and *white wine* ($N = 4898; d = 11; m \in \{10; 77\}$). The initial density is the multivariate Student $t$ distribution with $\nu = 10$ degrees of freedom, zero mean and covariance matrix $\Sigma_b$.

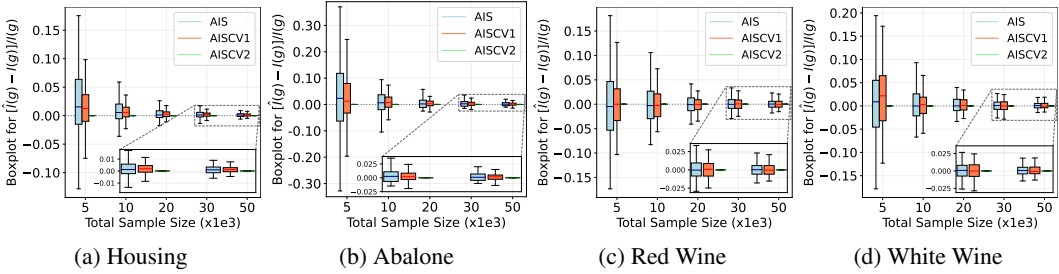

(a) Housing        (b) Abalone        (c) Red Wine        (d) White Wine

Figure 3: Bayesian linear regression: boxplots of $(\hat{I}(g) - I(g))/I(g)$ for $g(\theta) = \sum_{j=1}^d \theta_j^2$.

**Results.** Figure 3 presents the boxplots of the relative error $(\hat{I}(g) - I(g))/I(g)$, revealing the benefits of control variates even with polynomials of degree $Q = 1$. When $Q = 2$, the error of the AISCV2 estimate is virtually zero (see Table 3 in the supplement), in line with Proposition 2. The mean squared error of the AISCV1 estimate is smaller than that of the AIS estimate by a factor ranging between 2 and 10.

## 7 Discussion

While control variates are a well-known tool for Monte Carlo integration, standard methods do not allow the distribution of particles to evolve throughout the algorithm, as is the case for sequential methods. Within the standard adaptive importance sampling framework, we have developed a weighted least-squares procedure to improve numerical integration by incorporating control variates. The underlying adapted weights of this quadrature rule do not depend on the integrand and our non-asymptotic bound highlights the benefits of combining adaptive importance sampling with control variates. Different ways for constructing control variates are proposed. The method is fit for computing integrals with respect to the posterior density in Bayesian analysis, as the target density only needs to be known up to a multiplicative constant.

A limitation of the combined AISCV approach is that it requires the user to make quite some design choices, notably the sampling policy for the AIS part and the control variates for the CV part. These culminate into the factor $\tau$ in Assumption 3, which appears prominently in the error bound in Theorem 1 and which can be interpreted roughly as the standard deviation of $w\varepsilon$, where $w$ is the importance weight – well behaved when the policy is well-chosen in relation to the target density – and where $\varepsilon$ is some residual function – well behaved when the control variates are well-chosen with respect to the integrand. Further, choosing too many control variates may result in an ill-conditioned empirical Gram matrix or in overfitting. The least-squares solution could become unstable, requiring some kind of regularization, such as the LASSO [24].

## Acknowledgements

The authors are grateful to the Area chair and three anonymous Reviewers for their valuable comments and interesting suggestions. Aigerim Zhuman gratefully acknowledges a research grant from the *National Bank of Belgium*.

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
