# OpenReview forum: "A Quadrature Rule combining Control Variates and Adaptive Importance Sampling"
_NeurIPS.cc/2022/Conference — NeurIPS 2022 Accept_

### Official Review · Reviewer_KHDF · 2022-06-20

**Rating:** 7
**Confidence:** 4
**Soundness:** 4 excellent
**Presentation:** 4 excellent
**Contribution:** 3 good

**Summary:**

The authors present AISCV estimator that combined Adaptive Importance Sampling (AIS) and control variates (CV). The paper presents a theoretical proof of its convergence rates and presents practical considerations that are really helpful.

**Questions:**

Would it be worth having a baseline of just CV estimators? It is interesting to see how big the reduction is from CV -> AISCV. Especially, how would it compare on Figure 3. My thinking is the "virtually zero" variance is a CV effect rather than an AISCV effect. Is this correct?

**Limitations:**

The authors did not address this issue. I do not see any reason why the authors would need to address it either, the paper does not touch any sensitive area of society.

**Strengths And Weaknesses:**

The theoretical side of this paper is extremely well executed. Everything is carefully described and seems technically sound (I didn't carefully check details in the proof of Theorem 1, but seems valid based on what we know from AIS and CV. I did check the outline of the proof).

I would like the authors to reflect on if AISCV is just a combination of AIS and CV, or if they think it is greater than "the sum of its parts".

Could you Figure 1(a) and 1(b) have the same cutoff on the y-axis for easier visual comparison between input dimensions. Same where appropriate in Figure 2.

This paper has enough quality to be accepted to NeurIPS. The drawdown is, I think, that the short format of NeurIPS is making this paper poorer than it could be in a longer format. The experimental section is likely a bit sub-par to many other papers at this venue. Yet, the authors have more experiments in the appendix that I did not read carefully.

---

> ### Author Response · Authors · 2022-07-30
> **Response to Reviewer KHDF**
>
> We would like to warmly thank the reviewer for the valuable feedback, suggestions and the overall appreciation of our work, especially on the theoretical side.
>
> Concerning the general idea of combining AIS and CV: indeed, our methodology combines the two techniques in a more or less independent (although sequential) manner. Still, it is an interesting remark and challenge to seek for control variates that match the sampling policy well. A possible starting point is the parameter $\tau$ of Remark 1, which encodes the residual variance when combining the importance weights with the linear regression residuals. This quantity plays a central role in the theoretical analysis and makes the link between AIS and CV. Thank you for pointing out this direction of further research.
>
> Following your comment, the y-axes of the different figures now share the same cutoff value, facilitating visual comparison -- please check the rebuttal version.
>
> Concerning a baseline of just CV estimators in the numerical experiments: please note that a standard CV estimate requires access to samples from the target distribution. Given such access, it may not be relevant to use Adaptive Importance Sampling or Monte Carlo Markov chains. If we were to compute a CV estimate given the analytically available Gaussian posterior distribution, for example, we would probably take the family of Hermite polynomials since they form an orthonormal basis with respect to the Gaussian measure. Instead, our experiments are cast in a Bayesian spirit, the task being to compute an integral with respect to a posterior distribution from which one cannot sample directly.
>
> The reviewer is right concerning the "virtually zero" variance: this is a CV effect, the integrand lying close to its projection onto the linear space spanned by the control variates (see also Proposition 2).
>
> We thank the reviewer again for the constructive feedback, the positive appreciation, and the time and effort spent reviewing our work.

---

> > ### Comment · Reviewer_KHDF · 2022-08-08
> > **Thank you**
> >
> > I thank reviewers for their time in responding to my concerns and questions. I did not have many points that required improving, and therefore I remain positive about this paper. I suggest acceptance.

---

### Official Review · Reviewer_tPbu · 2022-07-06

**Rating:** 7
**Confidence:** 2
**Soundness:** 4 excellent
**Presentation:** 3 good
**Contribution:** 3 good

**Summary:**

This paper proposes a novel method for approximating intractable integrals that combines adaptive importance sampling and control variates. The authors prove a probabilistic bound on the error of their proposed estimator and demonstrate its superiority to the simple AIS estimator.

**Questions:**

Could it ever be the case that poorly (or even adversarially) constructed control variates result in unboundedly worse performance than AIS? Or is Assumption 2 a sufficient safeguard against this kind of behavior?

Can the authors speak to why their AISCV methods in Figure 2 demonstrate a higher variance than AIS in this setting? It seems odd to me that control variants, a method originally proposed as a variance reduction technique, could cause this behavior.

**Limitations:**

Related to my questions above, it seems that the use of control variates introduces a whole host of design choices that have the potential to result in poor performance. While the theoretical results speak to what requirements the control variates need to satisfy, in practice it seems like this could result in undesirable behavior if the requirements are not met.

**Strengths And Weaknesses:**

Overall, I found this paper an enjoyable read although I confess, many of the technical details surrounding the theoretical results escaped me.

1. In terms of originality, the primary proposed idea is not particularly inspired, in that it combines two commonly-used, existing methods.
2. The level of analysis, both theoretical and empirical, is compelling.
3. The paper is extremely well written and clear.
4. The work seems likely to be significant as the task of estimating integrals is commonplace and the proposed method appears sufficiently general so as to be widely applicable.

---

> ### Author Response · Authors · 2022-07-30
> **Response to Reviewer tPbu**
>
> We would like to sincerely thank the reviewer for the valuable feedback, the suggestions and the overall appreciation of our work. We address the different points below.
>
> The thought-provoking notion of 'adversarial' control variates, defined perhaps in terms of a min-max game and some appropriate rule, represents an interesting avenue for further research as (to the best of our knowledge) control variates nor AIS procedures have ever been compared under adversarial attacks. Still, in terms of asymptotic variance of the error term, it never harms to add linearly independent control variates (Assumption 2). In practice, however, choosing too many control variates may result in an ill-conditioned empirical Gram matrix and/or in overfitting. The OLS solution could become unstable, requiring some kind of regularization, such as the LASSO.
>
> Concerning the variance oddity in the boxplots of Figure 2, please note that it is a visual effect caused by the logarithmic scale. We apologize for the confusion. As expected, the AISCV methods do have a smaller variance than AIS as shown by the novel Figure 5 in the appendix drawn at a linear scale (please check the rebuttal revision). At a logarithmic scale, however, the smaller error values of the AISCV method (around 1e-5) look as if they are spread more widely than the larger error values of the AIS method (around 1e-3). For readability and clarity, Figure 2 now only presents the mean squared error, still on logarithmic scale.
>
> Regarding the choice of the control variates, the theoretical requirement in Assumption 2 is not hard so satisfy and is guaranteed for the construction methods in Section 5. Of course,  choosing control variates in some optimal way given the integrand and the target density can indeed be quite difficult, and it could very well occur that the obtained variance reduction does not justify the additional computational expenses.
>
> We thank the reviewer again for the positive feedback and for the time spent reviewing our paper.

---

> > ### Comment · Reviewer_tPbu · 2022-08-08
> > **Acknowledgement of response**
> >
> > Thanks for your detailed response! I appreciate the updated figures and the discussion about the choice of control variates and have increased my score to a 7.

---

### Official Review · Reviewer_KrYe · 2022-07-08

**Rating:** 7
**Confidence:** 3
**Soundness:** 3 good
**Presentation:** 3 good
**Contribution:** 3 good

**Summary:**

In this paper, the authors combine adaptive importance sampling (AIS) with control variates (CV) leading to a novel Monte Carlo estimator which they call AISCV. Non-asymptotic bounds were developed for the novel estimator and the experiments section demonstrated the superiority of AISCV versus a standard AIS estimate.

**Questions:**

Below I will list general questions/comments

1. In line 126, I think there is a typo. The paper says that $\mathbb{E}_f[g - \beta^\top h] = 0$ but it should be $\mathbb{E}_f[g - \beta^\top h] = \mathbb{E}_f[g]$.

**Limitations:**

There is no section in the paper describing the limitations of the approach. While presumably the limitations of AISCV are similar to AIS and CV, it would be beneficial to state them in the paper.

**Strengths And Weaknesses:**

# Strengths
I can not express how cool the proposed approach is. While standard CV estimators require $\beta$ to be recomputed for every target function $g$, AISCV sidesteps this completely! Instead---just like AIS---quadrature weights are computed using the particles and their corresponding importance weights, where, crucially, the quadrature weights **are not a function of** $g$! Thus, this process is done once and can be used for any $g$. Moreover, the bounds on the AISCV estimate are very impressive as well!

# Weaknesses

While the experiments section clearly demonstrated the superiority of AISCV, I wish a more practical experiment was done. AISCV was motivated by wanting to compute posterior expectations. Thus, it was disappointing that the only example done was Bayesian linear regression where the posterior is available analytically. Instead, it would have been more compelling if the AISCV estimate was compared against Bayesian linear regression, which can be sampled by using HMC for instance.

---

> ### Author Response · Authors · 2022-07-30
> **Response to Reviewer KrYe**
>
> We would like to sincerely thank the reviewer for the valuable feedback, enthusiasm and suggestions.
>
> The framework of Bayesian Linear Regression with real-world data was chosen so that the value of the integral of interest is analytically available when computing mean squared errors. Based on the suggestion, we have added in the supplement a state-of-the-art Monte Carlo Markov chain called NUTS (No-U-Turn Sampler), which is a self-tuning variant of Hamiltonian Monte Carlo (HMC).
>
> Thank you for spotting the typo: on l.126, it should indeed be $\mathbb{E}_f[g - \beta^\top h] = \mathbb{E}_f[g]$.
>
> A limitation of the combined AIS-CV approach is that it requires the user to make quite some design choices, notably the sampling policy for the AIS part and the control variates for the CV part. These culminate into the factor $\tau$ in Remark 1, which appears prominently in the error bound in Theorem 1 and which can be interpreted roughly as the variance of $w\epsilon$, where $w$ is the importance weight -- well behaved when the policy is well-chosen in relation to the target density -- and where $\epsilon$ is some residual function -- well behaved when the control variates are well-chosen with respect to the integrand. Further, choosing too many control variates may result in an ill-conditioned empirical Gram matrix and/or in overfitting. The OLS solution could become unstable, requiring some kind of regularization, such as the LASSO. If we are given the occasion, we will add a remark in the paper.
>
> We thank the reviewer again for the careful reading of our paper, the positive appreciation, and the interest in our research.

---

> > ### Comment · Reviewer_KrYe · 2022-08-08
> >
> > Thanks so much for your response! I must admit i made a typo in my initial review: I meant Bayesian **logistic** regression. I think an example like that would speak more to how this could affect real use cases!

---

### Meta-Review · Area_Chair_nUye · 2022-08-26

**Recommendation:** Accept
**Confidence:** Certain

**Metareview:**

This paper proposes a novel method to perform Monte Carlo integration combining control variates and annealed importance sampling. All reviewers agreed the algorithm was of interest, the theoretical evidence was strong, and the experimental results were sufficiently convincing, so there was a consensus on acceptance.

(As a minor aside, I would encourage the authors to consider the font sizes in their figures.)

**Award:**

No

---

### Decision · Program_Chairs · 2022-09-14

Accept